

# Identification and validation of an endoplasmic-reticulum-stress-related gene signature as an effective diagnostic marker of endometriosis

Tao Wang,  Mei Ji and  Jing Sun

Department of Gynecology, Shanghai Key Laboratory of Maternal-Fetal Medicine, Shanghai Institute of Maternal-Fetal Medicine and Gynecologic Oncology, Shanghai First Maternity and Infant Hospital, School of Medicine, Tongji Medical University, Shanghai, Pudong New Area, China

## ABSTRACT

**Background**. Endometriosis is one of the most common benign gynecological diseases and is characterized by chronic pain and infertility. Endoplasmic reticulum (ER) stress is a cellular adaptive response that plays a pivotal role in many cellular processes, including malignant transformation. However, whether ER stress is involved in endometriosis remains largely unknown. Here, we aimed to explore the potential role of ER stress in endometriosis, as well as its diagnostic value.

**Methods**. We retrieved data from the Gene Expression Omnibus (GEO) database. Data from the GSE7305 and GSE23339 datasets were integrated into a merged dataset as the training cohort. Differentially expressed ER stress-related genes (DEG-ERs) were identified by integrating ER stress-related gene profiles downloaded from the GeneCards database with differentially expressed genes (DEGs) in the training cohort. Next, an ER stress-related gene signature was identified using LASSO regression analysis. The receiver operating characteristic curve was used to evaluate the discriminatory ability of the constructed model, which was further validated in the GSE51981 and GSE105764 datasets. Online databases were used to explore the possible regulatory mechanisms of the genes in the signature. Meanwhile, the CIBERSORT algorithm and Pearson correlation test were applied to analyze the association between the gene signature and immune infiltration. Finally, expression levels of the signature genes were further detected in clinical specimens using qRT-PCR and validated in the Turku endometriosis database.

**Results**. In total, 48 DEG-ERs were identified in the training cohort. Based on LASSO regression analysis, an eight-gene-based ER stress-related gene signature was constructed. This signature exhibited excellent diagnostic value in predicting endometriosis. Further analysis indicated that this signature was associated with a compromised ER stress state. In total, 12 miRNAs and 23 lncRNAs were identified that potentially regulate the expression of *ESR1*, *PTGIS*, *HMOX1*, and *RSAD2*. In addition, the ER stress-related gene signature indicated an immunosuppressive state in endometriosis. Finally, all eight genes showed consistent expression trends in both clinical samples and the Turku database compared with the training dataset.

**Conclusions**. Our work not only provides new insights into the impact of ER stress in endometriosis but also provides a novel biomarker with high clinical value.

Corresponding author
Jing Sun, sunjing61867@tongji.edu.cn

# INTRODUCTION

Endometriosis, one of the most common benign gynecological diseases, is characterized by the abnormal ectopic presence and growth of endometrial-like tissue outside the uterus, predominantly affecting the pelvic cavity and ovaries (*Bulun et al., 2019*). Globally, up to 10% of reproductive-aged women suffer from this estrogen-dependent proinflammatory disease, which is accompanied mainly by chronic pelvic pain and infertility (*Bulun, 2009*). Generally, endometriosis negatively affects the quality of life of affected women and causes a substantial economic burden (*Soliman et al., 2016*). Given the lack of symptom specificity and understanding of its etiology, endometriosis is clinically challenging to manage, and definite diagnosis can be delayed by approximately 8–12 years (*Kiesel & Sourouni, 2019*). Current standard therapies for endometriosis include surgical lesion removal and hormone therapies, both of which have inevitable side effects and a high rate of recurrence (*Zakhari et al., 2021*). Thus, there is an urgent need to further elucidate the underlying molecular mechanisms involved in endometriosis initiation and progression, and to identify novel potential diagnostic markers and therapeutic targets.

The endoplasmic reticulum (ER), which resides in the cytoplasm of eukaryotic cells, is the main site for protein synthesis, processing, folding, structural maturation, and transport (*Fagone & Jackowski, 2009*). The ER possesses a tightly regulated enzymatic system to ensure that naïve proteins can be properly folded and modified, forming their unique three-dimensional structures before they traffic further into the cell (*Stein, 1975*). However, the ER appears to work near the limits of its protein-folding capability, and when burdened, accompanied by the accumulation of unfolded and misfolded proteins, an adaptive response called ER stress occurs (*Tabas & Ron, 2011*). In the context of ER stress, a conserved adaptive response called the unfolded protein response (UPR), which restores ER homeostasis, is initiated. The UPR is mediated by three ER transmembrane sensors: IRE1α (inositol-requiring enzyme 1α), pancreatic endoplasmic reticulum kinase (PERK), and ATF6 (activating transcription factor 6) (*Ron & Walter, 2007*; *Moncan et al., 2021*). Nevertheless, sustained ER stress can result in apoptosis (*Tabas & Ron, 2011*).

ER stress may regulate reproductive physiology and pathology (*e.g.*, endometriosis). During decidualization, endometrial stromal cells tend to secrete large amounts of proteins. This physiological process is accompanied by ER stress and subsequent UPR to prevent the accumulation of misfolded proteins (*Soczewski et al., 2020*). Meanwhile, in human endometrial cells, estrogen can inhibit ER stress, while progesterone can reverse this effect and activate ER stress, leading to cell apoptosis (*Choi et al., 2018*). Furthermore, ER stress activation is involved in the progesterone-mediated decrease in endometrial cell invasiveness (*Choi et al., 2019*). However, endometriosis has been reported to exhibit reduced ER stress intensity, which may be related to progesterone resistance (*Choi et al., 2019*). Likewise, inactive ER stress can no longer exert an inhibitory effect on

NF-κB activity in endometriotic stromal cells, leading to an increase in pro-inflammatory cytokine production (*Choi et al., 2021*). These results indicate that decreased ER stress responsiveness in endometriosis may favor ectopic lesion formation and growth. Several studies have consistently demonstrated the capacity of different compounds to induce endometriotic cell apoptosis by restoring and upregulating ER stress (*Park et al., 2018*; *Ryu et al., 2019*; *Park et al., 2017*). However, the exact molecular mechanism underlying the altered ER stress intensity in endometriosis remains largely unknown and requires further investigation.

Recent studies have highlighted the crucial role of immunological dysfunction in the pathogenesis of endometriosis. Endometrial fragments are thought to acquire the ability to evade immune surveillance and clearance following retrograde menstruation (*Ahn et al., 2015*). Thus, women who develop endometriosis are speculated to have defective immune systems. Many innate and adaptive immune factors have been verified to participate in the development of endometriosis. In many tumors, ER stress exhibits an immunosuppressive effect, causing malignant cells to have greater tumorigenic potential (*Cubillos-Ruiz, Bettigole & Glimcher, 2017*). However, whether ER stress modulates immune function in endometriosis has not been reported. Analysis of the relationship between the ER stress state and immune infiltration is helpful for further exploration of the underlying mechanisms of endometriosis.

In the present study, we aimed to explore the ER stress-related signature of endometriosis. We screened a combined merged cohort and successfully developed an eight-gene-based model, with a potential to accurately distinguish patients with endometriosis from healthy women. The diagnostic value of this model was verified using validation datasets and clinical specimens. Furthermore, a multifactorial lncRNA-miRNA-mRNA network was constructed to reveal the underlying regulatory mechanisms. In addition, the association between the ER stress signature and immune infiltration was evaluated using CIBERSORT. Overall, the in-depth bioinformatic findings from our work not only shed new light onto the pathogenesis of endometriosis but also provide novel potential diagnostic tools and candidate therapeutic targets.

## MATERIALS & METHODS

### Data sets and data collection

We retrieved the online GeneCards (https://www.genecards.org/) database to collect ER stress-related genes, and genes whose relevance score (defined as the relative relevance calculated by a practical scoring function between retrieved genes and ER stress) ≥ 7 were chosen for further analysis. Gene expression profiling of endometriosis was performed using the GEO (http://www.ncbi.nlm.nih.gov/geo) database. We downloaded five eligible datasets, including GSE7305, GSE23339, GSE51981, GSE105764, and GSE105765, all of which met the following criteria: (a) inclusion of a healthy control group; and (b) ten or more samples. Detailed information on all retrieved datasets is presented in Table 1. Among them, GSE7305 and GSE23339 were merged into an integrated cohort to serve as the training dataset, whereas GSE51981 and GSE105764 were set as external validation

Wang et al. (2024), *PeerJ*, DOI 10.7717/peerj.7070

Peer

**Table 1** Basic information of retrieved datasets.

| GEO | Data type | Platform | Samples | | | Control type | Disease type | rAFS stage | | Experiment type | Attribute |
|---|---|---|---|---|---|---|---|---|---|---|---|
| | | | Total | Control | EMS | | | I/II | III/IV | | |
| GSE7305 | mRNA | GPL570 | 20 | 10 | 10 | Normal[*] | Ovarian | – | – | array | training |
| GSE23339 | mRNA | GPL6102 | 19 | 9 | 10 | Normal[*] | Ovarian | – | – | array | training |
| GSE51981 | mRNA | GPL570 | 144[#] | 34 | 77 | Normal[*] | – | 27 | 48 | array | validation |
| GSE105764 | mRNA and lncRNA | GPL20301 | 16 | 8 | 8 | Eutopic[**] | Ovarian | 0 | 8 | RNA-seq | validation and ceRNA network construction |
| GSE105765 | miRNA | GPL11154 | 16 | 8 | 8 | Eutopic[**] | Ovarian | 0 | 8 | RNA-seq | ceRNA network construction |

**Notes.**

[*]Endometrium from disease-free women.

[**]Endometrium from women with endometriosis.

[#]GSE51981 contained other 37 samples with uterine abnormalities.

EMS, endometriosis; rAFS, revised American Fertility Society.

datasets. Furthermore, GSE105764 and GSE105765 containing the lncRNA and miRNA profiles, respectively, were used to construct the ceRNA network.

## Data processing and differentially expressed ER stress-related genes identification

First, the ComBat function in the "sva" R package was used to eliminate the batch effect in the training cohort. A gene expression box plot and PCA were used to verify the batch-normalized effect. Subsequently, "limma" and "DESeq2" R packages were employed to screen DEGs in microarray datasets (the merged training cohort and GS51981) and high throughput sequencing datasets (GSE105764 and GSE105765), respectively. Significant DEGs were selected with the cut-off criteria of adjusted $p$ value < 0.05 and absolute log fold change (FC) > 1. In addition, intersecting genes between DEGs in the training cohort and ER stress-related genes were defined as DEG-ERs for further analysis and model construction. Volcano plots and corresponding heat maps were generated to visualize the related results.

## Functional annotation analysis

GO and KEGG analyses were performed. First, the conversion from gene symbols to Entrez IDs was conducted using the "org.Hs.eg.db" R package. Then, we employed the "clusterProfiler" R package to conduct the above-mentioned functional enrichment analysis. A threshold $q$ value < 0.05 was set to define significantly enriched GO terms and signaling pathways.

## Protein–protein interaction network and ceRNA Network construction

The DEG-ERs identified were uploaded to the online STRING (https://string-db.org) database for protein-protein interaction (PPI) network construction. An interaction score >0.4 was selected as the filtering condition. Subsequently, the corresponding interaction files were downloaded for further analysis and visualization.

In addition to further exploring the regulatory mechanisms behind the constructed ER stress signature, we searched the miRNA 3.0 database (http://mirwalk.umm.uni-heidelberg.de/) for paired mRNA and miRNA interactions, and we retained only interactions that were also predicted in the miRDB database. The acquired miRNAs were filtered using the differential expression profile of the GSE105765 dataset. We further acquired the interactions between lncRNAs and final mapped miRNAs from StarBase v3.0 (http://starbase.sysu.edu.cn). Similarly, the lncRNA dataset GSE105764 was used to map the differentially expressed lncRNAs. The networks constructed in this study were further visualized using local Cytoscape software.

## ER stress signature construction and validation

The least absolute shrinkage and selection operator (LASSO) algorithm was used to construct an optimal ER stress-related gene signature for endometriosis using DEG-ERs. Based on the "glmnet" R package, the optimal values of the penalty parameter $\lambda$ were determined by 10-fold cross-validation. The quantification of the corresponding signature for each sample was defined as follows: ER stress score $= \sum_{i=1}^{N}(\text{coefi} \times \text{expri})$, where expri

is the relative expression of the genes in the signature, and coefi is the corresponding calculated LASSO coefficient of gene i. ROC analysis was performed to determine the diagnostic effectiveness of ER stress score using the "pROC" package in R. The area under the ROC curve (AUC) was used to estimate the diagnostic capability for discriminating endometriosis from control samples. The accuracy of the constructed model was validated using external datasets.

## Correlation between ER stress score and immunity

The CIBERSORT algorithm was used to evaluate the infiltration of 22 subsets of immune cells within the merged cohort. We then analyzed the correlation between different immune cell fractions and ER stress signatures.

## Reverse transcription-quantitative polymerase chain reaction (RT-qPCR)

Total RNA was extracted from ovarian endometriotic tissues ($n = 7$) and normal endometria of disease-free women ($n = 7$). Samples were placed in liquid nitrogen immediately after collection and subsequently transferred to $-80\,^\circ\text{C}$ for storage (within 1 h) for less than 3 months. 50 mg of each sample was used for RNA extraction after microdissection. RNA integrity was determined by nucleic acid gel electrophoresis. The concentration and purity of the total RNA in each sample was determined using a NanoDrop2000 spectrophotometer. Then, 1 μg qualified RNA (RIN ≥ 6) was reverse-transcribed into cDNA using the PrimeScript™ RT reagent kit (RK20429; ABclonal, Wuhan, China). Quantitative PCR was conducted using a QuantStudio5 instrument (Thermo Fisher Scientific, Waltham, MA, USA), three triplicates were performed, and intra-assay variation was below 1.5% to ensure repeatability. The relative expression levels of target genes were calculated using the $2^{-\Delta\Delta\text{CT}}$ method with β-actin as an internal control gene. Primers used in this study are listed in Table 2. The specificity of the primers was verified with primer blast (https://www.ncbi.nlm.nih.gov/tools/primer-blast/index.cgi?LINK_LOC=BlastHome) and melt curves (single peak). This study was approved by the Medical Ethics Committee of the Shanghai First Maternity and Infant Hospital (KS21198), and written informed consent was obtained from each patient. All tissues were collected at the proliferative stage, and none of the patients received any hormonal treatment for at least 3 months prior to surgery, the detailed information of clinical samples enrolled in this study was summarized in Table S1.

## Protein extraction and western blot assay

Total protein was extracted from endometriotic tissues ($n = 5$) and normal endometria ($n = 5$) using RIPA lysis buffer (PC101; EpiZyme, Shanghai, China). The total protein concentration was measured using the BCA protein assay (WB6501; NCM Biotech, Suzhou, China). Equivalent amounts of proteins (15 μg) were separated by 10% SDS-PAGE and transferred onto 0.45-μm polyvinylidene fluoride membranes (IPVH00010; Millipore, Darmstadt, Germany). After blocking, the membranes were incubated with primary antibodies against BOK (A21196, 1:1000; ABclonal, Woburn, MA, USA), PERK (A18196, 1:1000; ABclonal), p-PERK (AP0886, 1:1000; ABclonal), ATF6 (ab122897, 1:1500; Abcam,

**Table 2  Primers usud for qRT-PCR.**

| Gene | Forward or Reverse | Sequence (5′-3′) |
|---|---|---|
| *PTGIS* | Forward | GGGCCACACAGGGGAATATG |
| | Reverse | CGCTTGCCAAAGGATACTCTC |
| *ESR1* | Forward | CCCACTCAACAGCGTGTCTC |
| | Reverse | CGTCGATTATCTGAATTTGGCCT |
| *RYR2* | Forward | CATCGAACACTCCTCTACGGA |
| | Reverse | GGACACGCTAACTAAGATGAGGT |
| *AQP11* | Forward | ATCACCTTTTTGGTCTATGCAGG |
| | Reverse | TTGTATGGTTGTTATGCAGCCA |
| *APOA1* | Forward | CCCTGGGATCGAGTGAAGGA |
| | Reverse | CTGGGACACATAGTCTCTGCC |
| *BOK* | Forward | CAGTCTGAGCCTGTGGTGAC |
| | Reverse | TGATGCCTGCAGAGAAGATG |
| *HMOX1* | Forward | AAGACTGCGTTCCTGCTCAAC |
| | Reverse | AAAGCCCTACAGCAACTGTCG |
| *RSAD2* | Forward | CAGCGTCAACTATCACTTCACT |
| | Reverse | AACTCTACTTTGCAGAACCTCAC |
| *β-actin* | Forward | CATGTACGTTGCTATCCAGGC |
| | Reverse | CATGTACGTTGCTATCCAGGC |

Cambridge, UK), GRP78 (ab108615,1:1500; Abcam), and GAPDH (AB2100, 1:7000; NCM, Newport, RI, USA) at 4 °C overnight. The membranes were then incubated with secondary antibodies (G1213-100UL, 1:3000; Servicebio, Wuhan, China) at room temperature for 1 h. Finally, protein bands of interest were analyzed and quantified using ImageJ software.

## Turku database-based validation

The Turku database (https://endometdb.utu.fi/) is a web-based tool that integrates gene expression profiles and clinical information from endometriosis patients ($n = 115$) and normal control samples ($n = 53$) (*Gabriel et al., 2020*). This database was utilized to acquire gene expression patterns of the collected samples and validate the constructed ER stress signature.

## Statistical analysis

All statistical analyses were performed using R software (version 4.1.1). For continuous variables, Student's $t$-test (two groups) or one-way ANOVA (over two groups) was applied to assess intergroup differences when the data followed a normal distribution; otherwise, the Wilcoxon rank-sum test (two groups) or Kruskal–Wallis H test (over two groups) was used. For multiple comparisons, the Bonferroni correction method was used. For categorical variables, $\chi 2$ test was applied to estimate the differences among groups. All statistical analyses were two-sided and $p < 0.05$ indicated statistically significant.

## RESULTS

### Data processing and identification of differentially expressed ER stress-related genes

The workflow of this study is shown in Fig. S1. First, GSE7305 and GSE23339 were combined into a merged cohort that included 19 normal controls and 20 endometriosis samples. Boxplot analysis (Figs. 1A, 1C) and principal component analysis (PCA) (Figs. 1B, 1D) indicated that the batch effect was successfully eliminated. Moreover, the endometriosis group exhibited distinctive gene expression profiles from those of the control group (Fig. 1E), which further indicated the excellent quality of the merged cohort. A total of 958 differentially expressed genes (DEGs) were then identified, with 471 downregulated and 487 upregulated genes (Fig. 1F), and the top 10 upregulated and downregulated genes were further visualized using a heatmap (Fig. 1G). We acquired 785 ER stress-related genes from the GeneCards database (Table S2). A total of 48 differentially expressed ER stress-related genes (DEG-ERs) were both DEGs and ER stress-related genes (Fig. 1H). Among them, 19 were downregulated and 29 were upregulated (Fig. 1I). A protein–protein interaction (PPI) network was constructed to visualize their interactions (Fig. 1J).

### Functional annotation analysis

Functional annotation analysis was employed to further explore the functions of the DEG-ERs and reveal the pathways in which they may participate. Gene Ontology (GO) annotations comprised three aspects, namely the biological process (BP), cellular component (CC), and molecular function (MF). As a result, many ER stress-related terms were enriched. Furthermore, in the BP section, terms associated with the intrinsic apoptotic signaling pathway, calcium-mediated signaling, and epithelial cell apoptotic process were enriched (Fig. 2A), indicating that their perturbations may be mediators underlying the pathogenesis of endometriosis. In addition, DEG-ERs were mainly associated with ER lumen, blood microparticles, and platelet alpha granules in the GO-CC section (Fig. 2B). For GO-MF, molecular function activator activity, dystroglycan binding, and proteoglycan binding were significantly enriched (Fig. 2C). Kyoto Encyclopedia of Genes and Genomes (KEGG) analysis showed that the DEG-ERs were mainly involved in lipid and atherosclerosis, tumor necrosis factor signaling pathway, and cellular senescence (Fig. 2D).

### ER stress signature construction and external validation

An 8-gene signature was established in the training cohort using the least absolute shrinkage and selection operator (LASSO) regression algorithm (Figs. 3A, 3B). Among them, six genes (*PTGIS, RYR2, AQP11, APOA1, HMOX1,* and *RSAD2*) were upregulated in the endometriotic tissues, whereas the other two genes (*ESR1* and *BOK*) were downregulated (Fig. 3C). The ER stress score for each sample was determined as follows: ER stress score = $(1.327 \times PTGIS$ expression) + $(-0.076 \times ESR1$ expression) + $(0.950 \times RYR2$ expression) + $(0.607 \times AQP11$ expression) + $(0.172 \times APOA1$ expression) + $(-0.659 \times BOK$ expression) + $(0.543 \times HMOX1$ expression) + $(0.021 \times RSAD2$ expression). Receiver operating characteristic (ROC) curve analysis was performed to evaluate the effectiveness of the 8-gene signature for predicting endometriosis. As shown in Fig. 3D,

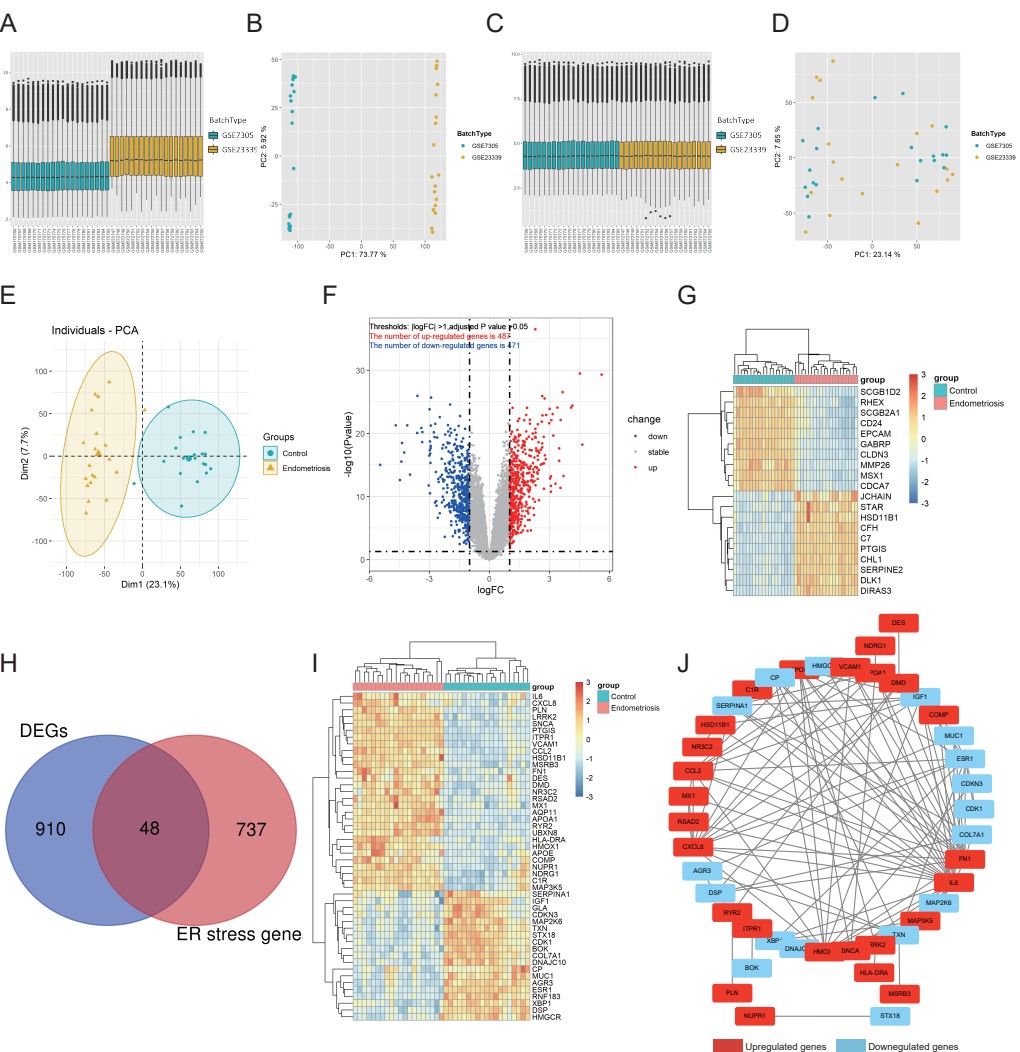

**Figure 1  Identification of DEG-ERs in the merged cohort.** (A–D) Boxplots and principal component analysis were employed to visualize batch correction efficacy before (A, B) and after (C, D) batch effect removal. (E) The data quality of the training cohort was visualized by PCA. (F) A volcano plot for DEGs. Blue and red dots denote significantly down-regulated and up-regulated genes, respectively, whereas grey dots represent genes with no significant difference. (G) A heatmap shows relative expression patterns of the top 10 up-regulated and down-regulated genes. (H) DEG-ER) identification based on intersecting genes between ER-stress-related genes and DEGs. (I) The final DEG-ERs were visualized using a heatmap. (J) The PPI network of the DEG-ERs. Each node represents a gene, while the lines show the interaction relationship among proteins encoded by the corresponding genes. Up-regulated and down-regulated genes are shown in red and blue, respectively. DEG-ER, differentially expressed ER stress-related gene; DEs, differentially expressed gene; PCA, principal component analysis; PPI, protein-protein interaction.

the area under the curve (AUC) in the combined dataset reached 1, indicating a strong diagnostic capability. Subsequently, we used two external cohorts to verify the accuracy of the constructed model. In the validation datasets, most of the gene expression patterns were consistent with those in the training cohort (Figs. 4A, 4D). The endometriosis group

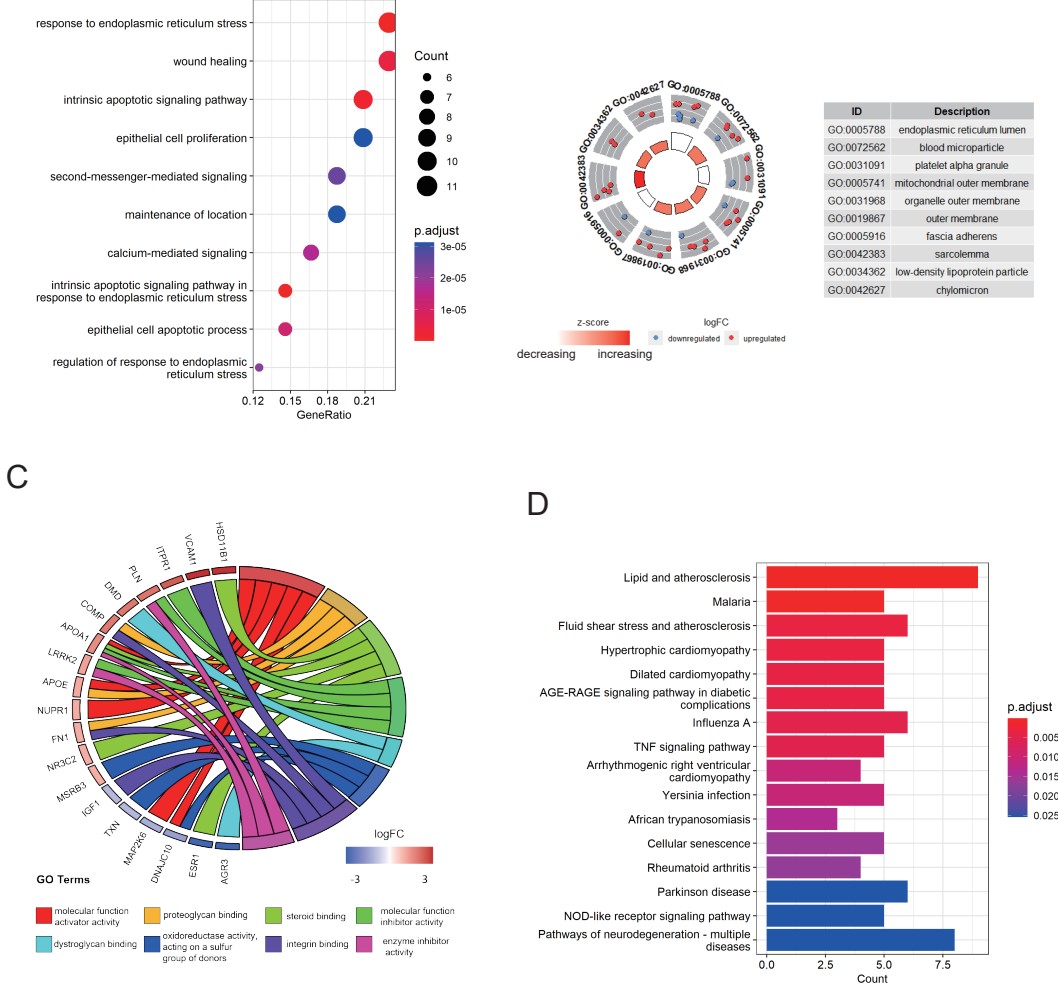

**Figure 2  GO and KEGG enrichment analyses of DEG-ERs in the merged cohort.** (A) Dot plot for the GO BP category. (B) Circle plot for the GO CC category. (C) Chord plot for the GO MF category. (D) Bar plot for the results of KEGG enrichment. GO, Gene Ontology; BP, biological process; CC, cellular component; MF, molecular function; KEGG, Kyoto Encyclopedia of Genes and Genomes.

had a higher ER stress score than the control group (Figs. 4B, 4E). Furthermore, the AUC value was 0.749 (95% CI [0.652–0.846]) (Fig. 4C) in the GSE51981 dataset, whereas it was equal to 1 in the GSE105764 cohort (Fig. 4F). Overall, these results suggest the predictive ability of the ER stress-related signature in exclusive diagnosis of women patients with endometriosis cases from disease-free women.

Additionally, given the well-established impact of the menstrual cycle on gene expression in the endometrium, and both the training set and validation set GSE51981 contained proliferative-stage and secretory-stage samples (Table S3), we performed a differential analysis adjusted for menstrual cycle stages. And the subgroup analyses of the training

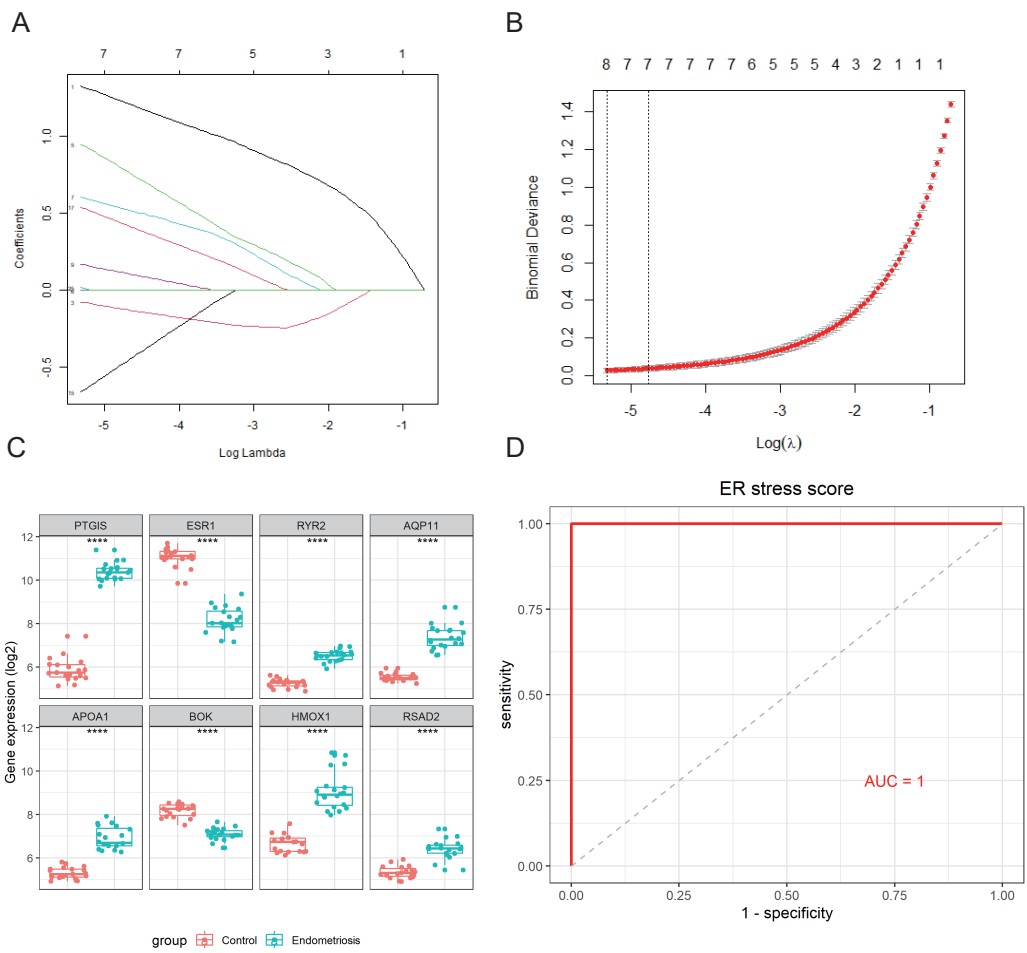

**Figure 3   Identification of ER stress-related gene signature in endometriosis.** (A) LASSO coefficient profiles of DEG-ERs. (B) Selection of the optimal lambda value in the LASSO model. (C) The expression levels of the eight genes in the constructed signature. ****$p < 0.0001$. (D) ROC analysis for the calculated ER stress score. LASSO, least absolute shrinkage and selection operator; ROC, receiver operator characteristic; AUC, area under the curve.

set and the validation set GSE51981 based on the menstrual cycle demonstrated the stage-independent effectiveness of our results (Fig. S2).

## Endometriosis exhibited an inactivated ER stress state

The activation of ER stress can be marked by a set of related intracellular proteins, such as PERK (encoded by *EIF2AK3*), ATF6, GRP78 (encoded by *HSPA5*), and XBP1. To determine the amount of ER stress occurring in endometriosis, we analyzed the differential expression levels of these canonical molecules in the training and validation cohorts. We found that most of these markers were downregulated in the endometriosis group, indicating that the ER stress signaling pathway was inhibited in endometriosis (Figs. 5A–5C). Furthermore, the ER stress score we constructed was negatively correlated with most of these markers (Figs. 5D–5F), indicating that higher scores correlated with a more inactive ER stress state.

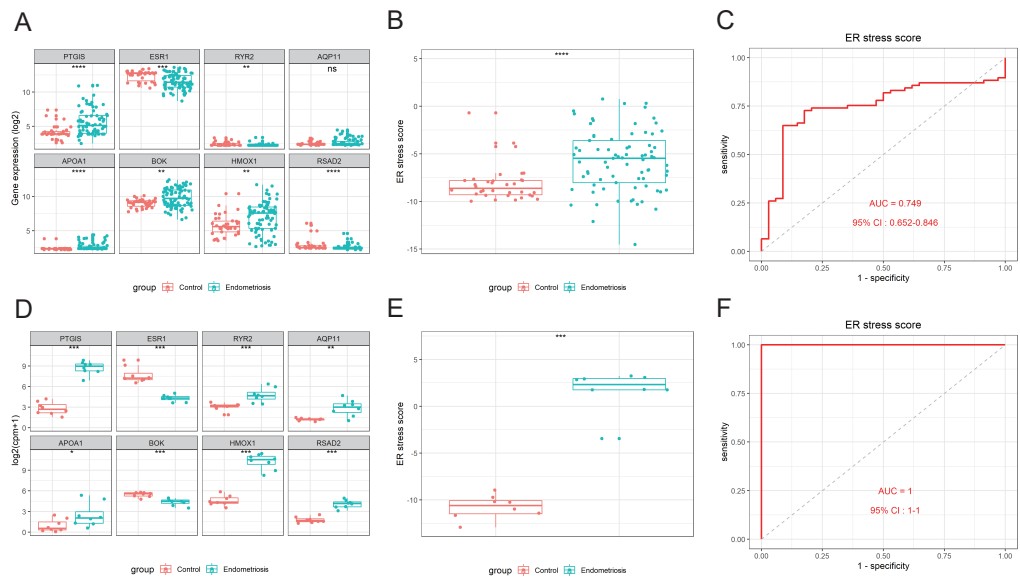

**Figure 4** **External validation of the ER stress-related gene signature.** (A, D) The differential expression levels of the eight signature genes in GSE51981 (A) and GSE105764 (D). (B, E) The relative ER stress scores of endometriosis samples and normal control samples in GSE51981 (B) and GSE105764 (E). (C, F) ROC analysis of the calculated ER stress score in GSE51981 (C) and GSE105764 (F). ns, not significant, $*p < 0.05$, $**p < 0.01$, $***p < 0.001$; ROC, receiver operator characteristic; AUC, area under the curve.

Using clinical samples, we demonstrated that the protein levels of PERK, p-PERK, ATF6, GRP78, and BOK were decreased in endometriosis using clinical samples (Figs. 5G, 5H).

## Multifactorial network construction and pathway analysis

A multifactorial interaction network was constructed to explore the potential regulatory mechanisms underlying perturbation of the ER stress signature, a multifactorial interaction network was constructed. A total of 465 mRNA-miRNA interactions were identified in the miRwalk 3.0 database based on the canonical interaction relationship in the 3′UTR regions. We obtained 12 mRNA-miRNA pairs under the criteria that the miRNAs were differentially expressed in the GSE105764 dataset and had opposite expression levels compared with the corresponding mRNAs (Table S4). Then, StarBase v3.0 database was utilized to acquire the miRNA-lncRNA interactions. Similarly, only lncRNAs that overlapped with the differentially expressed lncRNAs in the GSE105765 dataset and exhibited the same expression trends as the corresponding mRNAs were retained (Table S5). A total of 30 miRNA–lncRNA pairs were identified. Based on these results, we successfully constructed a ceRNA network to partially delineate the regulatory mechanisms of the ER stress signature in endometriosis (Fig. 6).

## Correlation between gene signature and infiltrated immune cells in endometriosis

First, the CIBERSORT algorithm was applied to analyze the infiltration of 22 distinct immune cells in the training cohort. The immune composition of each sample is shown in

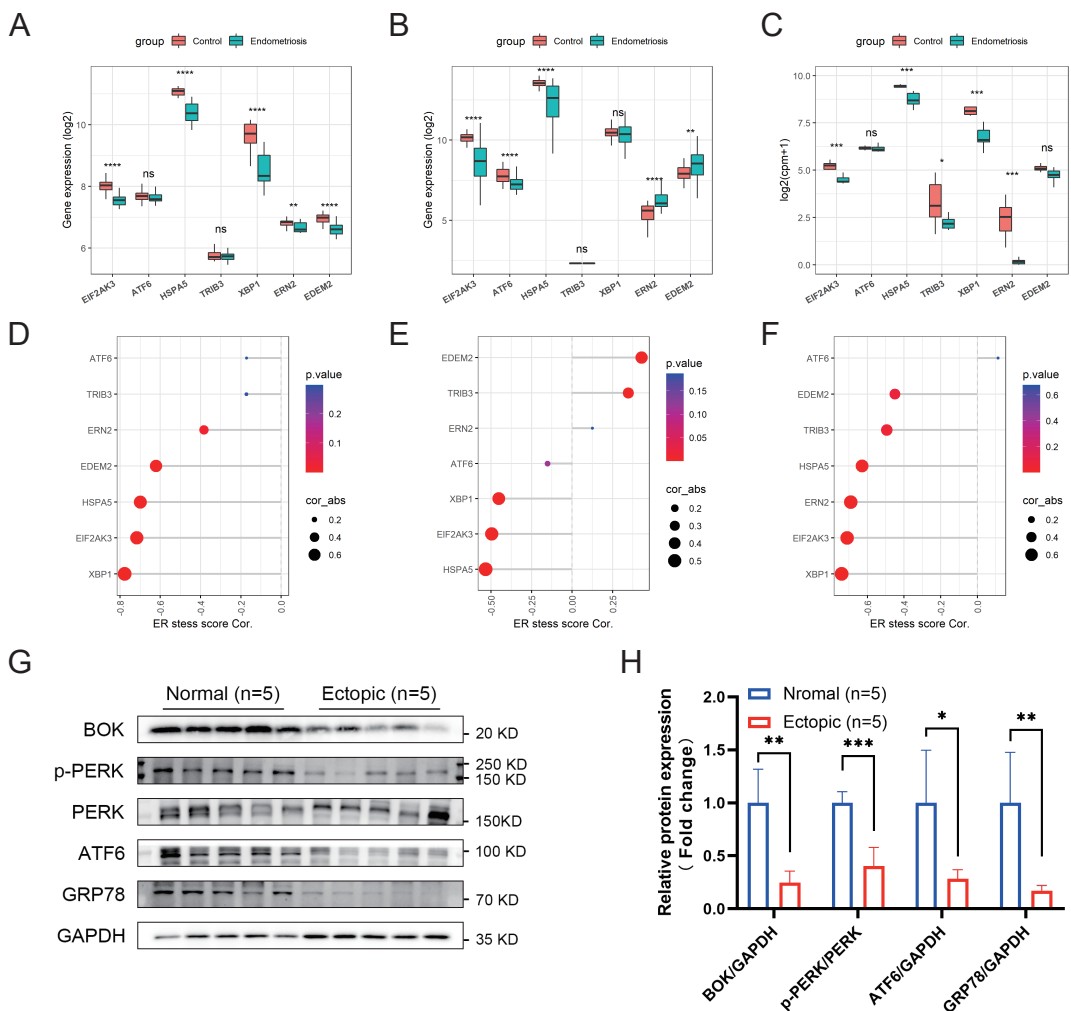

**Figure 5   Association between the ER stress signature and the corresponding ER stress intensity.** (A–C) The differential expression levels of canonical ER stress markers in the merged cohort (A), GSE51981 (B), and GSE105764 (C). (D–F) Correlation between the ER stress score and different ER stress markers. The sizes of the dots denote the strength of the correlation between, while the color of the dots represents the *p*-value. (G) The protein levels of ER stress markers detected by western blotting in clinical samples. The blots were cut prior to hybridization with antibodies to save the amount of antibody used. (H) Quantification analysis of blots using ImageJ software. ns, not significant, *$p < 0.05$, **$p < 0.01$, ***$p < 0.001$.

Fig. 7A. Endometriosis indeed showed a different immune infiltration pattern compared with that of the normal controls (Fig. 7B), indicating that immune factors may contribute to the pathogenesis of endometriosis. We further analyzed the differences in infiltration levels between endometriosis and control samples, as presented in Fig. 7C. Ten significantly differentially infiltrated immune cells were identified, including CD4 memory activated T cells, gamma delta T cells, activated mast cells, resting memory CD4 cells, M2 macrophages, activated dendritic cells, T follicular helper cells, NK cells, activated CD8 T cells, and regulatory T cells (Tregs) (Fig. 7D). Thereafter, we evaluated the correlation between the ER stress signature and 10 differentially infiltrated immune cells. The results revealed

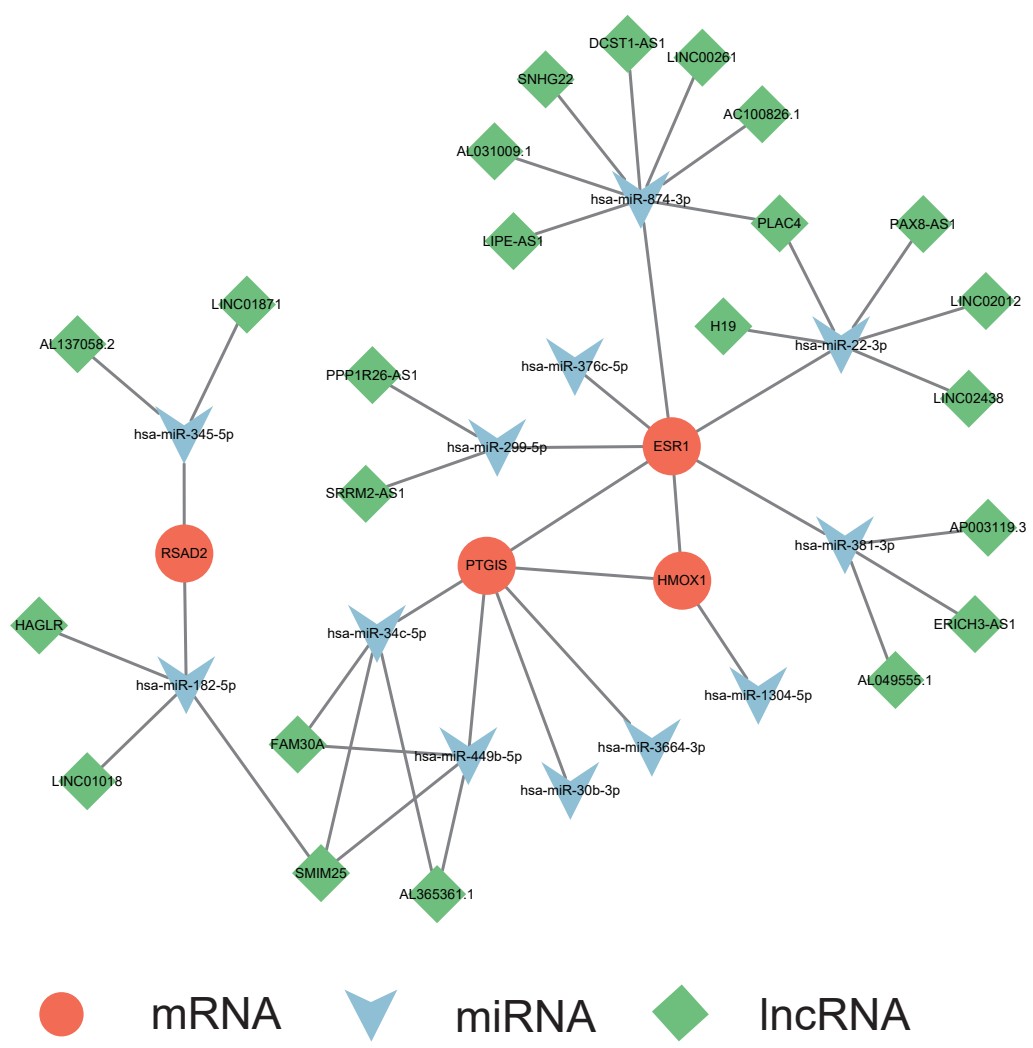

**Figure 6** Construction of mRNA-miRNA-lncRNA multifactorial regulatory network in endometriosis. miRNA, microRNA; lncRNA, long noncoding RNA.

that the constructed gene signature was predominantly positively correlated with M2 macrophages and showed a strong negative relationship with NK cell activation (Figs. 7E, 7F).

## Reverse transcription-quantitative polymerase chain reaction results of genes in the signature

To validate the expression levels of the eight genes in the constructed ER stress signature, we conducted reverse transcription-quantitative polymerase chain reaction on clinical specimens. The expression patterns of *PGTIS*, *ESR1*, *RYR2*, *AQP11*, *APOA1* and *BOK* were consistent with those in the training dataset ($p < 0.05$) (Figs. 8A–8F). Although the differences in *HMOX1* and *RSAD2* between the endometriotic tissues and normal controls were not statistically significant, their expression trends were in line with
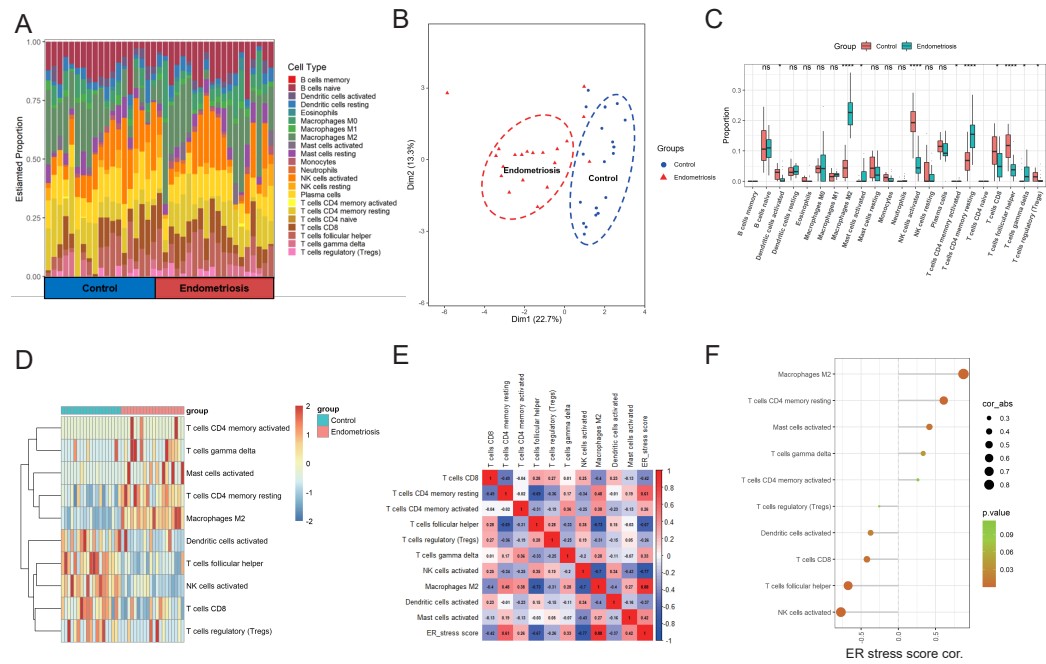

**Figure 7  Correlation between ER stress signature and immune characteristics in the merged cohort.**
(A) A stacked bar chart showed the abundance of 22 immune cells for each sample. (B) PCA was employed to determine the differences in immune characteristics between endometriotic tissues and normal control tissues. (C) The box plot shows the differential infiltrating levels of 22 immune cell types between endometriotic tissues and normal control tissues. (D) The 10 differentially infiltrating immune cells were visualized using a heatmap. (E) Correlation matrix of the ER stress score and the 10 differentially infiltrating immune cells. (F) Correlation between the ER stress score and the 10 differentially infiltrating immune cells. The sizes of the dots denote the strength of the correlation between the ER stress score and the immune cells, while the color of the dots represents the $p$-value. ns, not significant, $*p < 0.05$, $**p < 0.01$, $***p < 0.001$.

expectations (Figs. 8G, 8H). Additionally, the endometriotic tissues had higher ER stress scores, as calculated using the aforementioned formula (Fig. 8I). Furthermore, the online endometriosis database Turku was used to further evaluate the reliability of our results. Although there was no statistical confirmation, we found that the expression trends of all eight genes corroborated those in the training cohort, especially for ovarian lesions (Fig. S3).

## DISCUSSION

Endometriosis affects approximately 200 million women worldwide (*Rogers et al., 2009*). The course of endometriosis can last most of the patients' lifespan, as the disease process can be initiated during the first menses, persisting until menopause. Although significant efforts have been dedicated to revealing the exact pathogenesis of endometriosis, a unified, convincing conclusion has not yet been reached. A recently published literature review shows that numerous studies have focused on the important role of ER stress in endometriosis (*Al-Hetty et al., 2023*). In our study, we successfully constructed an

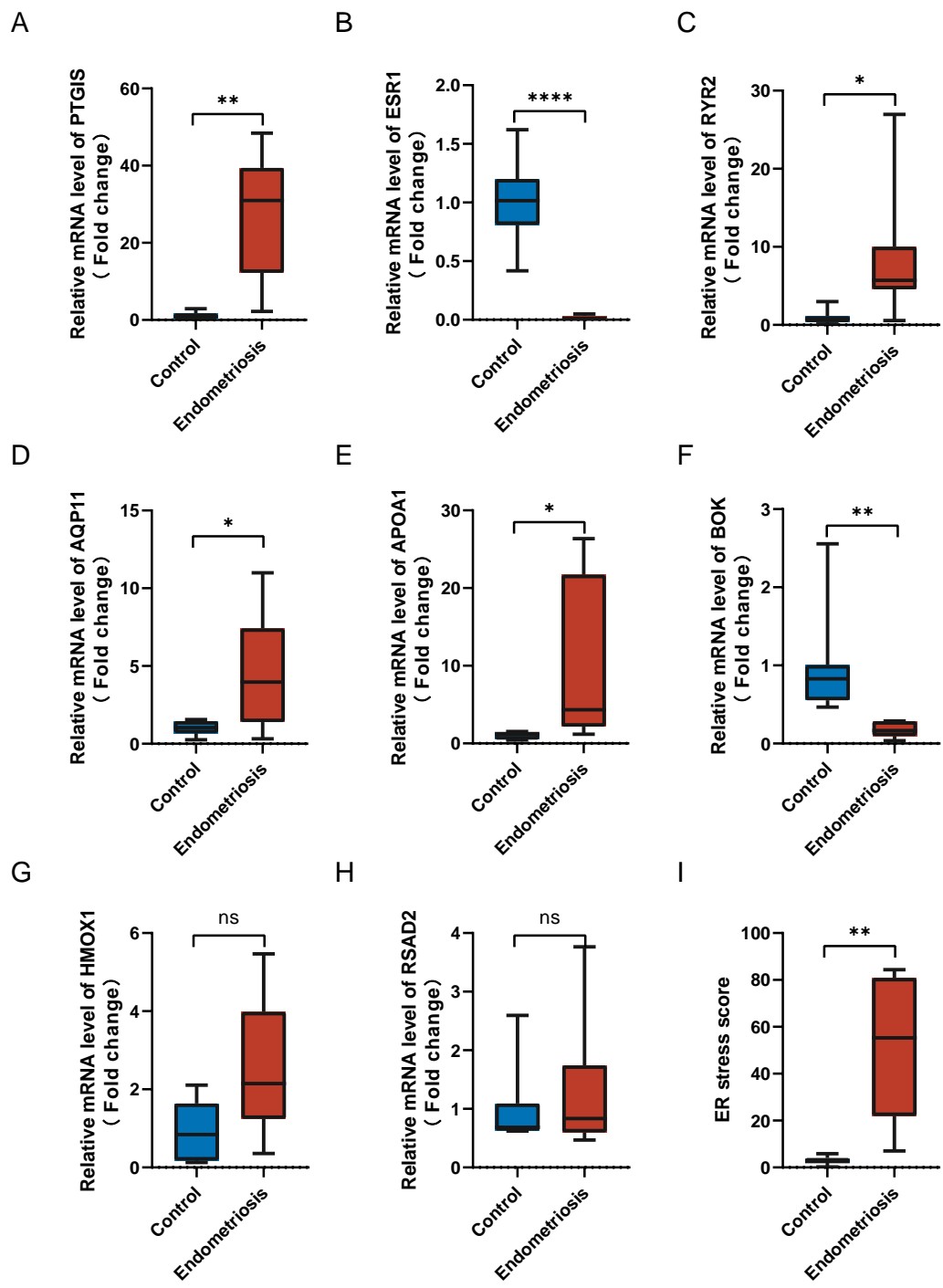

**Figure 8** **Validation of signature genes in clinical samples.** (A–H) The relative differential mRNA levels of *PTGIS*, *ESR1*, *RYR2*, *AQP11*, *APOA1*, *BOK*, *HMOX1*, and *RSAD2* between the endometriotic tissues and control tissues. (I) The calculated ER stress scores of the endometriotic tissues and control tissues. ns, not significant, *$p < 0.05$, **$p < 0.01$, ***$p < 0.001$.

ER stress-related signature, which performed well in discriminating endometriosis from normal samples. Additionally, in-depth analysis revealed that this 8-gene model was correlated with decreased ER stress intensity and might affect immune characteristics within the endometriosis microenvironment.

We identified 48 DEG-ERs in the training cohort. Functional annotation analysis indicated that these genes may play a predominant role in regulating ER stress and apoptotic processes in endometriosis. In general, these two signaling pathways are highly interconnected, and failure to adapt to persistent cellular ER stress will trigger apoptosis (*Hetz, 2012*). As previously mentioned, several compounds have been shown to suppress endometriosis by inducing apoptosis in an ER stress-dependent manner. Additionally, DEG-ERs can also affect wound healing, a process involved in migration and invasiveness, which plays a critical role in endometriosis. ER stress is activated by progesterone, which reduces the invasiveness of normal endometrial stromal cells (*Choi et al., 2019*). However, progesterone resistance is considered a hallmark of endometriosis (*Patel et al., 2017*); thus, increased endometriotic stromal cell invasiveness may partially originate from blunt ER stress reactivity to progesterone (*Choi et al., 2019*). These results suggest that ER stress may contribute to endometriosis and serve as a potential therapeutic target. However, considering that ER stress participates in many biological processes, such as decidualization, whether these drugs could affect the physiological systemic homeostasis of ER stress requires further investigation. Hence, development of drugs that target the activation of ER stress in endometriosis may avoid unnecessary side effects.

By performing LASSO analysis, we successfully developed an ER stress-related prediction model in the training cohort; six genes (*PTGIS, RYR2, AQP11, APOA1, HMOX1,* and *RSAD2*) were upregulated, whereas the other two (*ESR1* and *BOK*) were downregulated. PTGIS catalyzes the conversion of prostaglandin H2 to prostacyclin (prostaglandin I2), which is a potent vasodilator and inhibitor of platelet aggregation. In insulin-producing cells, PTGIS overexpression attenuates cytokine toxicity by suppressing ER and mitochondrial stress-mediated cell death pathways (*Gurgul-Convey & Lenzen, 2010*). In addition, forced PTGIS expression has been reported to promote the macrophage switch to the M2 phenotype (*Pan et al., 2021*). Considering the pivotal role of M2 in endometriosis, we speculated that elevated PTGIS may also exert its function by influencing macrophage polarization. Compared with normal endometrial stromal cells, endometriotic stromal cells express extraordinarily lower ESR1 and significantly higher ESR2 levels (*Yang et al., 2015*). This aberrantly low ESR1:ESR2 ratio changes the mode of action of estrogen and influences diverse pathological processes in endometriosis, including apoptosis, proliferation, invasiveness, and inflammation (*Yilmaz & Bulun, 2019*). In estrogen receptor α (ERα)-positive breast cancer, estrogen induces the rapid anticipatory activation of the UPR *via* ERα (*Andruska et al., 2015*). RYR2 is one of the components of the calcium channel that participates in maintaining cellular $Ca^{2+}$ homeostasis (*Cui et al., 2017*). RYR2 downregulates the PERK signaling pathway, thus attenuating ER stress in TM-induced cardiomyocytes (*Zeng et al., 2020*). Meanwhile, RYR2 depletion perturbed cardiomyocyte maturation, which was linked to the drastic activation of ER stress pathways

(*Guo et al., 2022*). In the uterine arteries, pregnancy can increase both RYR1 and RYR2 protein levels, leading to elevated $Ca^{2+}$ spikes and spontaneous transient outward currents (STOCs) (*Hu et al., 2020*). However, under high-altitude hypoxic conditions, these effects were suppressed by enhanced ER/oxidative stress (*Hu et al., 2020*). AQP11 belongs to a third subfamily of aquaporins, namely, unorthodox or "superaquaporins" (*Ishibashi, Tanaka & Morishita, 2021*). AQP11 has been reported to ameliorate ER stress by maintaining intracellular $H_2O_2$ homeostasis (*Tanaka et al., 2016*; *Frühbeck et al., 2020*).

APOA1 is one of the main components of high-density lipoprotein (HDL) and has been identified as a novel potential biomarker for a wide variety of tumors (*Moore et al., 2006*). APOA1 has been reported to reduce ER stress in hepatocytes by modifying lipid transport (*Liu et al., 2014*; *Guo, Zhang & Wang, 2017*). Additionally, human chorionic gonadotropin (hCG) treatment was able to decrease APOA1 levels in disease-free baboons, whereas APOA1 was upregulated by hCG administration in endometriotic baboons (*Sherwin et al., 2010*). This inverse response may partly explain the endometriosis-related implantation failure. BCL-2 ovarian killer (BOK) belongs to the BCL2 family and functions as a strong pro-apoptotic regulator. Intriguingly, BOK has recently been identified as a selective modulator of ER stress-triggered apoptotic pathways. In $BOK^{(-/-)}$ cells, the ER stress-induced mitochondrial apoptotic response was blunted, whereas no differences were observed in response to other apoptotic stimuli (*Carpio et al., 2015*). Thus, downregulation of BOK in endometriosis might render endometriotic cells resistant to ER stress-induced apoptosis. Heme oxygenase-1 (HO-1, encoded by *HMOX1*) is a ubiquitous enzyme that physiologically catalyzes heme metabolism, accompanied by the production of carbon monoxide (CO) (*Loboda et al., 2016*). HO-1 and its products protect cells against various stimuli, including ER stress. HO-1/CO signaling has been demonstrated to negatively regulate ER stress in endothelial cells in an acute lung injury model (*Kim et al., 2007*; *Chen et al., 2018*). In endometriosis, HO-1 is up-regulated (*Allavena et al., 2015*) and the functional polymorphism of *HMOX1* is associated with endometriosis (*Milewski et al., 2021*). RSAD2 has been identified as a highly inducible ER protein with antiviral activity (*Qiu, Cresswell & Chin, 2009*), but its other functions have rarely been reported. RSAD2 has also been identified as a prognostic predictor in triple-negative breast cancer, but the exact mechanism remains unclear (*Jiang et al., 2016*).

ER stress and the resulting UPR can be induced by a variety of events, such as oxidative stress, starvation stimulation, and calcium homeostasis disruption (*Xu, Bailly-Maitre & Reed, 2005*). The UPR is a determinant of cell fate under ER stress, and all three UPR sensors monitor the level of misfolded proteins and attempt to restore cellular homeostasis under stress conditions so that cells can function normally. In the presence of irremediable ER stress, such as high or chronic ER stress, the UPR pathway is sustained, ultimately inducing cell death (*Shore, Papa & Oakes, 2011*). Pelvic hypoxic conditions are the predominant challenge for retrograded endometriotic cells. However, hypoxia stimulates the UPR and ER stress-mediated apoptosis (*Delbrel et al., 2018*). Nevertheless, in endometriosis, downregulation of ER stress intensity has been observed. In line with previous reports (*Choi et al., 2019*), we found that most ER stress markers were downregulated in endometriosis, suggesting compromised ER stress status. As mentioned
above, in our constructed signature, the expression of *PTGIS*, *RYR2*, *AQP11*, *APOA1*, and *HMOX-1*, which could attenuate ER stress activity, was elevated, whereas the mediator (*ESR1*) and downstream effector (*BOK*) of ER stress were downregulated. This indicated the potential role of signature genes in maintaining low ER stress intensity in endometriosis. Indeed, the calculated ER stress score was negatively correlated with most canonical ER stress markers. Based on these results, we speculated that decreased susceptibility to ER stress and the downstream apoptotic process experienced by endometriotic cells could favor disease progression.

In addition, the eight-gene model performed well in discriminating endometriosis from disease-free women in the training dataset and two external validation cohorts, indicating remarkable diagnostic value. And subgroup analyses stratified by menstrual cycle stage verified cycle stage-independent effectiveness of our constructed ER stress-related signature. The differences in the AUC values of ER stress score in the two validation cohorts might result from the differences in sample sizes (111 and 16 samples were included in our study from GSE51981 and GSE105764, respectively). In this study, we experimented with a smaller sample size and a larger number of validation data. Thus, we were assured that the model and the identified genes are generalizable in a bigger cohort. The expression levels of the eight genes were further validated in clinical samples and the online Turku Endometriosis Database. Although our results provide a novel diagnostic tool with high discrimination capacity, a biopsy should still be performed to assess the expression of ER stress markers in the ectopic endometrium, which may limit the clinical transformation of our study. Considering the extensive alterations of methylation modifications in endometriosis (*Dyson et al., 2014*), we hypothesized that the altered RNA expression of these signature genes might be due to the altered methylation status. In the future, we will further evaluate the methylation changes in signature genes in circulating cell-free DNA in the disease state, which may aid in the development of noninvasive diagnostic methods.

To explore the mechanisms underlying the ER stress-related signature in endometriosis, we constructed a ceRNA network. The predicted miRNAs and lncRNAs were mapped to differentially expressed miRNAs in GSE105765 and differentially expressed lncRNAs in GSE105764. Finally, four mRNAs, 12 miRNAs, and 23 lncRNAs were selected to construct a multifactorial regulatory network. Among them, many regulatory associations have been revealed; for instance, miR-22 was demonstrated to inhibit ESR1 expression in breast cancer (*Vesuna et al., 2021*), whereas H19 could act as a sponge of miR-22 in many cell types (*Gan, Lv & Liao, 2019*; *Sun, Mao & Ji, 2021*). However, further functional experiments are required to confirm these regulatory axes in endometriosis.

Recently, substantial data has indicated that dysregulation of the immune system contributes to the pathogenesis of endometriosis. By analyzing 22 distinct levels of immune cell infiltration, we found that endometriosis exhibited distinctive immune characteristics. Among them, the abundance of the 10 immune fractions were significantly different. In various tumors, ER stress bestows malignant cells with immune-evasion capacity; however, whether the ER stress response impedes the development of anti-endometriosis immunity remains unknown. We performed a correlation analysis between ER stress scores and

10 differentially infiltrated immune subtypes to reveal the potential immunoregulatory effects of ER stress in endometriosis. The ER stress score was positively correlated with macrophages M2, CD4 memory resting T cells, and activation of mast cells, T cells, and gamma delta, and negatively correlated with activated NK cells, follicular helper T cells, CD8 T cells, and activated dendritic cells. Among these, several types have been reported to influence the progression of endometriosis. M2 macrophages have been widely documented to play pivotal roles in immunosuppression and neuroangiogenesis during endometriosis (*Wu et al., 2017*). Mast cells (MCs) are known to mediate allergic reactions, which are activated mainly in an IgE-dependent manner. MCs are natural components of the human uterus, and their activation participates in normal menstruation regulation by promoting endometrial shedding (*Menzies et al., 2011*). Compared to normal endometrium, endometriotic lesions recruited a greater number of MCs (*Anaf et al., 2006*). Furthermore, in an experimental rat endometriosis model, the activation of MCs was demonstrated to mediate the estrogen growth promotion effect on endometriotic lesions (*Lin et al., 2015*). MC inhibitors or stabilizers have potential therapeutic value in treating endometriosis (*Binda, Donnez & Dolmans, 2017*), but their exact efficacies need to be further confirmed in standardized clinical trials. NK cells act as guards against foreign and harmful objects, including viruses and tumors. It is reasonable to speculate that NK cells are involved in the clearance of retrograde endometrial debris, whereas the decreased cytotoxicity of pelvic NK cells in endometriosis could allow endometriotic cells to survive and successfully implant (*Du, Liu & Guo, 2017*). Overall, correlation analysis suggests that altered ER stress intensity may impede normal immune reactions in endometriosis; however, further studies are needed to verify this hypothesis.

Although some published studies have also attempted to explore the underlying molecular mechanisms, immune microenvironment, and potential diagnostic markers of endometriosis using bioinformatics methods (*Geng et al., 2022*; *Sun, Gan & Sun, 2022*), our study is the first bioinformatics-based study to explore the relationship between ER stress and endometriosis, which provides novel insights into the pathogenesis of endometriosis. However, our study also has some limitations. First, this study was conducted based on public databases, and the sample size for clinical validation is relatively small. We will further expand our validation cohort in the future. Second, the expression and diagnostic value of the eight genes at the protein level requires additional investigation. Third, the tissue type and disease grade varied among the different datasets (including the Turku database), which could introduce bias in estimating the diagnostic value of the constructed model. Finally, the association between the eight signature genes and ER stress intensity was reported in non-endometrial/endometriotic cell lines, and subsequent confirmatory experiments in endometrial/endometriotic cells are required.

## CONCLUSIONS

In conclusion, we successfully constructed an ER stress-related signature for endometriosis, which performed well in predicting endometriosis in the training cohort and external validations. Additionally, most genes in this model were correlated with reduced ER

stress intensity, and we assume that low ER stress activity may enable endometriotic cells to be resistant to the corresponding apoptotic events induced by pelvic stimuli, such as hypoxic conditions. Further analysis revealed that the ER stress signature is associated with dysregulated immune function in endometriosis. Collectively, our work revealed the potential role of ER stress in the pathogenesis of endometriosis and the underlying regulatory mechanisms and provides an excellent tool to aid in the diagnosis of endometriosis. In our future work, we will further explore how these genes in the model affect the ER stress intensity and progression of endometriosis in a context-dependent manner.

**List of abbreviations**

| | |
|---|---|
| **ER** | Endoplasmic Reticulum |
| **GEO** | Gene Expression Omnibus |
| **DEGs** | Differentially Expressed Genes |
| **DEG-ERs** | Differentially Expressed ER Stress-Related Genes |
| **GO** | Gene Ontology |
| **KEGG** | Kyoto Encyclopedia of Genes and Genomes |
| **PPI** | Protein–protein Interaction |
| **LASSO** | The Least Absolute Shrinkage and Selection Operator |
| **ROC** | Receiver Operating Characteristic Curve |
| **AUC** | Area Under the Curve |

## ACKNOWLEDGEMENTS

We would like to thank Editage for English language editing.

### Funding

This work was supported by grants from the Science and Technology Commission of Shanghai Municipality (22Y11906100) to Jing Sun and Shanghai Outstanding Academic Leaders Plan to Jing Sun (Year 2019). The funders had no role in study design, data collection and analysis, decision to publish, or preparation of the manuscript.

### Grant Disclosures

The following grant information was disclosed by the authors:
The Science and Technology Commission of Shanghai Municipality: 22Y11906100.
Shanghai Outstanding Academic Leaders Plan (Year 2019).

### Competing Interests

The authors declare there are no competing interests.

## Author Contributions

- Tao Wang conceived and designed the experiments, performed the experiments, analyzed the data, prepared figures and/or tables, authored or reviewed drafts of the article, and approved the final draft.
- Mei Ji analyzed the data, prepared figures and/or tables, and approved the final draft.
- Jing Sun conceived and designed the experiments, authored or reviewed drafts of the article, and approved the final draft.

## Human Ethics

The following information was supplied relating to ethical approvals (i.e., approving body and any reference numbers):

The Ethics Committee of Shanghai First Maternity and Infant Hospital

## Data Availability

The R codes for this study are available in the Supplementary Files. The sequencing data that support the findings of this study are publicly available at GEO: GSE7305, GSE23339, GSE51981, GSE105764, and GSE105765.

## Supplemental Information

Supplemental information for this article can be found online at http://dx.doi.org/10.7717/peerj.17070#supplemental-information.

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
