# Peer review of "Identification and validation of an endoplasmic-reticulum-stress-related gene signature as an effective diagnostic marker of endometriosis"

_PeerJ, doi:10.7717/peerj.17070_

## Round 0.1 · original submission · Major Revisions

Most of the comments identified for consideration by the reviewers are relatively straightforward for you to address.

The one key exception to this is the issue of the menstrual cycle raised by reviewer-2. I agree that this is an absolutely critical issue that needs to be addressed. Hence, your paper will need further experimental analysis to address this issue, and this may in turn impact the results. Hence, I regard this as a major revision and note that careful and rigorous addressing of this issue is essential for further progression with this study.

Reviewer 1 ·

Basic reporting

1. Paper is very well written and easy to follow.

Experimental design

No comment

Validity of the findings

No comment

Additional comments

In this manuscript authors have aimed to explore the potential role of ER stress in endometriosis, as well as its diagnostic value. They have retrieved data from the Gene Expression Omnibus (GEO) database which was used as training dataset. Differentially expressed ER stress-related genes (DEG-ERs) were identified by integrating ER stress-related gene profiles downloaded from the GeneCards database with differentially expressed genes (DEGs) in the training cohort. To evaluate their potential ER-markers for clinical diagnostic value, they have calculated ROC, to check the sensitivity of the proposed (constructed) model using the Differentially expressed ER-stress markers. They have also evaluated these markers by QRT and western blot.
Comments for Authors:
1. Paper is very well written and easy to follow. Authors have also discussed the limitations of their study. But there are several similar published studies (e.g. https://pubmed.ncbi.nlm.nih.gov/36532015/) and authors should emphasize how this study is different and what else this study contribute further compared to other published data.
2. Please provide some more information about the dataset being used for the study. For example, the dataset used were microarray data or single cell RNA seq or just RNAseq! Similarly, provide information about Turku database.
3. Why the cutoff value for DEGs is 1 (so low)? In the text (line 234-36), BOK has been included in both upregulated and downregulated genes. Please correct it in the text/
4. It was confusing as to why ceRNA comparative analyses were done when its role in ER-stress was not pursued further to show it as causative factor for ER-stress.
5. WB and QRT (fig 5 and Fig8) should be placed together to show differences in gene and protein expression (in my opinion). Also, were the same samples used for QRT and WB?

Reviewer 2 ·

Basic reporting

Wang et al.'s study of identification of ER stress-related genes associated with endometriosis explores the intriguing connection between endometriosis and genes involved in ER stress. The manuscript provides valuable insights into potential molecular mechanisms contributing to endometriosis, drawing on publicly available gene expression datasets and employing robust experimental methodologies.

The manuscript is well-written content and comprehensive experimental design. The authors effectively leverage publicly available gene expression datasets, conduct pathway analyses, and validate key genes and proteins using qPCR and Western blot techniques. The overall study design demonstrates a thoughtful and systematic approach to unraveling the genetic underpinnings of endometriosis.

Experimental design

However, a notable concern arises from the omission of menstrual cycle information in the analysis. Given the well-established impact of the menstrual cycle on gene expression in the endometrium and estrogen related genes, it is crucial to consider this factor in the interpretation of results. The menstrual cycle stages can significantly influence gene expression patterns, potentially confounding the findings if not appropriately addressed.

To enhance the study's robustness, I recommend to incorporate menstrual cycle information from the samples in analyses. Performing a differential expression analysis adjusted for menstrual cycle stages would provide a more accurate representation of gene expression changes specifically associated with endometriosis, minimizing potential confounding effects.

Validity of the findings

While the manuscript demonstrates a commendable exploration of ER stress-related genes, a critical examination reveals gaps in the collection and inclusion/exclusion criteria of clinical samples.

One prominent concern centers around the lack of detailed information regarding the process of sample collection. Comprehensive reporting of sample collection methodologies is imperative in ensuring the reliability and reproducibility of the study. The omission of critical details, such as patient demographics, clinical histories, and specific collection protocols, raises questions about the study's external validity and potential bias in sample selection.

Reviewer 3 ·

Basic reporting

Submitted manuscript concerns an emergent problem of endometriosis. The authors made huge analysis of expression data and propose a gene signature for diagnosis. The work is well documented, including informed consent forms, MIQE guidelines form, R code and several additional data within supplementary materials. However after careful reading of the manuscript, I pin point out some inaccuracies which should be clarified.

Experimental design

Review

Introduction:

What is the link between cancers and reproductive physiology (line 73-77)? Why is it important in your study? Please clarify this.

Materials and methods:

Which p-value correction method was used?

RT-qPCR validation:
Please provide a clinical data of validation cohort (control + endometriosis)
Line 177: “ß-actin as a normalizer” it should be “ ß-actin as an internal control gene”.

Results:

Lines 243-250: There are some mixed up figures. It seems that Figure 4C describes two different graphs. Additionally, the results of validation cohorts are very different. The ER stress score from the cohort GSE51981 doesn’t differentiate normal samples from endometriosis samples opposite to the cohort GSE105765. Please clarify this.

Graphs of Figure 4 are mixed up. It should be (A,C), (B,D), (C,E).
Where is fig. 4F?

Figure 8. Better way to show relative gene expression results is to use box and whisker plot not bar plot.


Discussion:

Line 380-381: Please, improve English. To many “however”

Line 40-402: The authors plans to investigate circulating cell free DNA in the future, however what would be the relationship between RNA signature and cfDNA. Please clarify this.
The differences between cohorts GSE51981 and GSE105765 should be discussed much more deeper.

Validity of the findings

This work is well validated however some data should be clarified.

Additional comments

No additional comments.

---

## Round 0.2 · Minor Revisions

Thanks for attending to the revisions. Reviewer-2 has a few little points to tidy up on an annotated file in their review and recapitulated here for info; the numbers refer to original review.

1. If so, I suggest to remove this sentence to not mislead the reader of undefined relation between
cancer and endometriosis. Please stay focused on reproductive physiology.
2. Thank you for the explanation.
3. Correct.
4. Very good.
5. See point 11.
6. The Figure 4 is correct.
7. As above.
8. Now is much more suitable.
9. O.K.
10. Thank you for clarification.
11. Thank you for this additional discussion, however I wish to see (in the supplementary files) a figure
- box plot with whiskers, just like Fig, 4B but separate for proliferative and secretory phase.
12. Very good.

Thank you for addressing all concerns.

Reviewer 2 ·

Basic reporting

no comment

Experimental design

no comment

Validity of the findings

no comment

Additional comments

The authors have addressed all my comments from the previous submission.

Reviewer 3 ·

Basic reporting

Very well reported research.

Experimental design

Good.

Validity of the findings

Well validated

Annotated reviews are not available for download in order to protect the identity of reviewers who chose to remain anonymous.

---

## Round 0.3 · accepted · Accept

Thanks - I am happy with these changes and the response about Figure S2.
Congratulations.